# The Impact of Prolonged Inflammation on Wound Healing

**DOI:** 10.3390/biomedicines10040856

**Published:** 2022-04-06

**Authors:** Judith C. J. Holzer-Geissler, Simon Schwingenschuh, Martin Zacharias, Johanna Einsiedler, Sonja Kainz, Peter Reisenegger, Christian Holecek, Elisabeth Hofmann, Barbara Wolff-Winiski, Hermann Fahrngruber, Thomas Birngruber, Lars-Peter Kamolz, Petra Kotzbeck

**Affiliations:** 1Research Unit for Tissue Regeneration, Repair and Reconstruction, Division of Plastic, Aesthetic and Reconstructive Surgery, Department of Surgery, Medical University of Graz, 8036 Graz, Austria; judith.geissler@medunigraz.at (J.C.J.H.-G.); elisabeth.hofmann@joanneum.at (E.H.); lars.kamolz@medunigraz.at (L.-P.K.); 2COREMED-Cooperative Centre for Regenerative Medicine, Joanneum Research Forschungsgesellschaft mbH, 8010 Graz, Austria; einsiedler.johanna@gmail.com; 3HEALTH-Institute for Biomedicine and Health Sciences, Joanneum Research Forschungsgesellschaft mbH, 8010 Graz, Austria; simon.schwingenschuh@joanneum.at (S.S.); sonja.kainz@joanneum.at (S.K.); peter.reisenegger@joanneum.at (P.R.); christian.holecek@joanneum.at (C.H.); thomas.birngruber@joanneum.at (T.B.); 4Diagnostic and Research Institute of Pathology, Medical University of Graz, 8010 Graz, Austria; martin.zacharias@medunigraz.at; 5Division of Endocrinology and Diabetology, Department of Internal Medicine, Medical University of Graz, 8036 Graz, Austria; 6AKRIBES Biomedical GmbH, 1030 Vienna, Austria; barbara.wolff-winiski@akribes-biomedical.at (B.W.-W.); hermann.fahrngruber@gmx.at (H.F.)

**Keywords:** prolonged inflammation, wound healing, inflammation, resiquimod R848, wound model

## Abstract

The treatment of chronic wounds still challenges modern medicine because of these wounds’ heterogenic pathophysiology. Processes such as inflammation, ischemia and bacterial infection play major roles in the progression of a chronic wound. In recent years, preclinical wound models have been used to understand the underlying processes of chronic wound formation. However, the wound models used to investigate chronic wounds often lack translatability from preclinical models to patients, and often do not take exaggerated inflammation into consideration. Therefore, we aimed to investigate prolonged inflammation in a porcine wound model by using resiquimod, a TLR7 and TLR8 agonist. Pigs received full thickness excisional wounds, where resiquimod was applied daily for 6 days, and untreated wounds served as controls. Dressing change, visual documentation and wound scoring were performed daily. Biopsies were collected for histological as well as gene expression analysis. Resiquimod application on full thickness wounds induced a visible inflammation of wounds, resulting in delayed wound healing compared to non-treated control wounds. Gene expression analysis revealed high levels of IL6, MMP1 and CD68 expression after resiquimod application, and histological analysis showed increased immune cell infiltration. By using resiquimod, we were able to show that prolonged inflammation delayed wound healing, which is often observed in chronic wounds in patients. The model we used shows the importance of inflammation in wound healing and gives an insight into the progression of chronic wounds.

## 1. Introduction

Treating chronic wounds has been a challenge in modern medicine for years. The annual treatment costs of chronic wounds are estimated to be 25 billion dollars in the US alone [1,2]. A breakthrough in the treatment of chronic wounds is still missing, not because of a lack of research or funding, but rather because of the complexity and diversity in the aetiology of chronic wounds. Many chronic wounds differ in their pathophysiology and arise from multiple different underlying diseases, such as diabetes mellitus, peripheral artery disease or chronic venous insufficiency [3,4]. These diseases lead to a lack of oxygen and nutrients, or increased inflammation in the skin and are intrinsic causes of chronic wounds, but extrinsic factors, such as wound infections or biofilm-forming bacteria colonisation, worsen the chances of healing [5].

Prolonged inflammation plays a major role in defective wound healing and in the development of chronic wounds [6]. Inflammation is an integral part of the physiological phases of wound healing, which aims to purge the wound of cell debris and potential pathogens [3]. Neutrophils and macrophages, which are attracted through chemotaxis, clear the wound of bacteria. Once the wound is devoid of pathogens and cell debris, neutrophils are disposed of by apoptosis without causing further tissue damage or aggravating inflammation. The remaining macrophages release various growth factors that activate the surrounding keratinocytes, fibroblasts and endothelial cells [3,4,7]. If the leucocytes are hindered from infiltrating or clearing the wound or are unable to be disposed of by apoptosis, the process of entering the proliferation phase in wound healing is delayed or inhibited. Through pro-inflammatory stimuli, the wound is kept in a cycle of inflammation that is difficult to break [3].

Prolonged inflammation and ischemia-mediated lack of oxygen and nutrients, combined with bacterial colonisation or infection, are the cellular factors that substantially influence the formation of a chronic wound [3]. Most of the established preclinical wound models do not take prolonged inflammation into consideration, so they create wounds with delayed wound healing that do not correctly reflect chronic wounds in patients. Additionally, convenient models using rodents or rabbits are preferred because of easy handling, but do not consider the known differences in skin structure or immune response compared to humans [8,9]. Since prolonged inflammation plays a prominent role in delayed wound healing, we aimed to investigate inflammation induced by topical resiquimod application in wounds, and the effects of resiquimod-induced inflammation on wound healing. Resiquimod is a member of the imidazoquinolinamines and an agonist to Toll-like receptor 7 (TLR7) and TLR8. Resiquimod induces production of interferon α, tumour necrosis factor α (TNFalpha)and other cytokines [10,11]. Similar to its predecessor, imiquimod, it is used for topical treatment of skin lesions, such as herpes simplex warts, actinic keratosis or cutaneous T-cell lymphoma [12,13,14], but with local inflammation as a side effect of treatment [12]. Besides treatment, imiquimod is frequently used to mimic inflammatory skin disorders, such as psoriasis, in rodents [15]. Toll-like receptors play an important role in wound healing, as they are the first line response in the defence against invading pathogens [16]. There are currently 10 members of the human TLR family characterized that recognize different viral as well as bacterial components and mediate an intracellular response [17,18]. Although most of the TLR responses and pathway activations promote physiological wound healing, an over-expression can decelerate these processes or even have opposite effects, such as prolonged inflammation and extensive scarring. An increase in TLR7 was observed in the wounds of diabetic patients with wound healing disorders [19,20,21].

Our aim was to assess the influence of topical resiquimod application on inflammation and wound healing. Furthermore, we wanted to investigate the inflammatory response to resiquimod in wounds by molecular and histological analysis.

## 2. Materials and Methods

### 2.1. Preparation of the Resiquimod R848 Solution

Resiquimod R848 (tlrl-r848, InvivoGen Europe, Toulouse, France) was dissolved in sterile endotoxin-free water to generate a 5 mL resiquimod stock solution (15.9 mM). The working solution of resiquimod consisted of 593 µL deionised water, 7.5 µL Tween 80 (9005656, Sigma-Aldrich Handels GmbH, Vienna, Austria), 750 µL propylene glycol (82280, Sigma-Aldrich Handels GmbH, Vienna, Austria), and 135 µL of the resiquimod R848 stock solution (1.4 mM). The working solution was freshly prepared in a 2 mL tube (Biosphere^®^ SC Micro Tube 2.0 mL, Sarstedt, Nümbrecht, Germany) and applied to 4 wounds per day. All solutions used were medical grade.

### 2.2. Animal Study

Ten female domestic pigs (weight approximately 15 kg) were purchased from a certified breeder and acclimatised for at least one week before starting the experiment. One day before the start of the experiment, the backs of the pigs were shaved. On Day 1, the pigs were anaesthetised using 0.5 mg/kg midazolam (Midazolam Accord, Accord Healthcare Limited, Devon, UK), 10 mg/kg ketamine (Ketasol, Graeub, Bern, Switzerland), 2 mg/kg azaperone (Stresnil, Elanco GmbH, Bad Homburg vor der Höhe, Germany) and 0.2 mg/kg butorphanol (Butomidor, Richter Pharma AG, Vienna, Austria). Each pig received 6 full thickness wounds (3 × 3 cm) on its back. Onto four of these wounds, 300 µL of the 1.4 mM resiquimod R848 working solution was applied daily with a syringe for 6 days. The remaining 2 wounds served as untreated control wounds. Control and resiquimod-applied wounds were randomly arranged on the back of each pig to avoid a location bias (Figure 1A). After the last day of resiquimod application (Experimental day 6), the wounds were examined for another 10 days before the pigs were euthanised at day 16 of the experiment (Figure 1B). During the entire experiment, the pigs received 25 mg/h fentanyl patches for analgesia and had free access to food and water. The study was approved by the Austrian Ministry of Education, Science and Research (BMBWF-66.010/0198-V/3b/2018).

### 2.3. Wound Dressing Changes

Wound dressings were changed daily. The pigs were awake during dressing changes and were restrained in a hammock. The wounds were covered with a layer of Mepilex^®^lite (Mölnlycke Health Care, Göteborg, Sweden) and fixed with Tegaderm™Film (3M, Maplewood, MN, USA) and Leukotape^®^K (BSN medical, Hamburg, Germany). Additionally, every day after the removal of the wound dressing, the wounds were photo-documented and scored by the same person based on a visual scoring scale.

### 2.4. Visual Wound Scoring

The overall wound score was divided into an inflammation score and an area score. The following parameters were included in the inflammation score: presence of pus, swelling or erythema and the width of the erythema if present. For each parameter, individual numbers were given based on the phenotype of the wound, as listed in Appendix A. Briefly, pus (score 0–2), erythema (score 0–4), erythema width (score 0–1) and swelling (score 0–2) were scored, with a maximum possible sum score of 8 for the inflammation score. Representative pictures of wounds and the scoring scheme are shown in Appendix A. The following parameters were included in the area score: wound size, moisture of the wound, presence of granulation tissue, scabbing and necrosis. For each parameter, individual numbers were given based on the phenotype of the wound, as listed in Appendix A. Briefly, moisture of the wound (score 0–1), size (score 0–3), granulation tissue (score 0–2), scab (score 0–3) and necrosis (score 0–4) were assessed, with a maximum possible score of 13 and a minimum possible score of 0 for the area score. Representative pictures of wounds and the scoring scheme are shown in Appendix A. Adding these two scores resulted in the overall wound score, with a maximum possible score of 21 points and a minimum possible score of 0 points. Full thickness excision wounds started with a total score of 7, with 0 points for the inflammation score. This scoring system has been previously developed at JOANNEUM RESEARCH and is used routinely in studies where wound healing is investigated.

### 2.5. Gene Expression Analysis

Excision and punch biopsies (6 mm) were snap-frozen on dry ice and were stored at −80 °C until further processing. Skin biopsies were homogenized in Qiazol using Magnalyzer Beads and the Magnalyzer instrument. Immediately after homogenisation, RNA was isolated using an RNAeasy^®^ Lipid Tissue Mini Kit (Qiagen, Hilden, Germany), according to the standard protocol. RNA concentration was measured on a Nanodrop (Thermo Fisher Scientific, Waltham, MA, USA) instrument, and measurements were performed in duplicate. One microgram of total RNA was used for cDNA synthesis, which was performed using the iScript^TM^ gDNA clear kit (Biorad, Hercules, CA, USA). All qPCRs were performed using gene primers of the selected target genes (IL8, IL10, COX-2, CD68, TGFb, INFg) and the SsoAdvanced^TM^ Universal SYBR^®^ Green Supermix (Biorad, Hercules, CA, USA). For IL6 and MMP1 detection, gene-specific Real-Time PrimePCR Assays (Biorad, Hercules, CA, USA) and the TaqManGene Expression Master Mix (Thermo Fisher Scientific, Waltham, MA, USA) were used. YWHAZ served as the housekeeping gene. qPCR reactions were run on a Biorad CFX384 cycler (Biorad, Hercules, CA, USA). Relative expression was calculated using the Pfaffl method [22].

### 2.6. Histology

For histological analysis, punch biopsies were taken from the centre of each wound on the euthanised pigs. Skin biopsies were fixed in 10% neutrally buffered formalin solution and paraffin was embedded, according to standard procedures. Three micrometre thick sections were prepared and attached to charged glass slides (Menzel Superfrost Plus, Thermo Fisher Scientific, Waltham, MA, USA), and stained with haematoxylin and eosin. Images were captured with an Aperio ScanScope AT digital slide scanner (Leica Biosystems, Wetzlar, Germany) at 40-fold magnification.

### 2.7. Quantification of Immune Cells

All haematoxylin- and eosin-stained tissue slides were evaluated for an abundance of lymphocytes, plasma cells, neutrophils and eosinophils within the dermis. Cell counting was performed on a microscope with an ocular field diameter of 0.5 mm on 400× magnification (Nikon MICROPHOT-FXA, Nikon, Tokyo, Japan), representing 1 high power field (HPF). For each case, 5 randomly chosen HPFs within the dermis were scored, after which the counts of each cell type were added up, leading to the respective cell counts/mm^2^. Counting and data acquisition were performed by a trained pathologist, blinded to group labels. Slides with air inclusions or similar problems had to be excluded from the analysis.

### 2.8. Open Flow Microperfusion and Cytokine Measurements in Dermal Wounds

Dermal interstitial fluid (dISF) was collected from wounded skin and unwounded control areas with dermal open flow microperfusion (dOFM) [23,24,25]. dOFM probes were implanted into the dermis of wounded and unwounded skin areas on each pig’s back. Forty-eight hours before dOFM sampling, wounds with a depth of 0.25 mm were inflicted with a dermatome and treated with resiquimod cream (1 g/wound at d1 and 0.5 g at d2, with 1.4 µmol resiquimod/g cream). The implanted dOFM probes were perfused with a physiological isotonic solution (ELO-MEL isoton, Fresenius Kabi Austria GmbH, Graz, Austria) containing 2% human serum albumin (Kedrion S.p.A., Barga, Italy), with a flow rate of 1 µL/min. After a run-in phase of about 15 min, we collected samples of dISF for 2 h, resulting in 120 µL dISF per dOFM probe. Per pig, we used 6 dOFM probes in control wounds and 6 dOFM probes in resiquimod-treated wounds. In total, 4 wounded areas and 4 control sites were analysed in two pigs. For cytokine quantification, dISF was analysed in duplicate by using 96-well 4-Spot Prototype (MSD) Porcine 4-plex (P/N: N75ZA-1) (Meso Scale Discovery, Rockville, MD, USA).

### 2.9. Data Analysis and Statistics

Data were transferred and calculated by using Microsoft Excel 2016 (Microsoft Corporation, Redmond, WA, USA), and statistical analyses were performed with GraphPad Prism 9 (GraphPad Prism version 9.3.1 for Windows, GraphPad Software, San Diego, CA, USA, www.graphpad.com, accessed on 1 March 2022). Figures were made using GraphPad Prism 9 and Inkscape 0.92 (Inkscape Project. (2020). Inkscape. Retrieved from https://inkscape.org, accessed on 1 March 2022). Data are presented as mean ± standard deviation (SD). Two-way ANOVA was used, and corrected for multiple comparisons using the Sidak method. For comparison of two study groups, an unpaired *t*-test with a Welch’s correction was used. A mixed-model analysis and Geisser Greenhouse correction, followed by a Sidak correction, was used for not normally distributed data; *p* ≤ 0.05 was considered statistically significant.

## 3. Results

### 3.1. Topical Resiquimod Application Increased Wound Size and Delayed Wound Healing

To investigate the effects of prolonged inflammation on wound healing, full thickness wounds in pigs were induced with the TLR7/8 agonist resiquimod.

Visual assessment of wounds showed that resiquimod application on full thickness wounds led to prolonged inflammation and slower wound healing when compared to untreated control wounds. The untreated control wounds showed normal stages of wound healing, granulation tissue and re-epithelialisation with hardly any scab left at the end of the experiment (Figure 2A top row). Resiquimod-induced wounds had little granulation tissue, but formed a thick necrotic scab that remained firmly on top of the wounds until the end of the experiment (Figure 2A bottom row).

Visual scoring was used to quantify wound area score and wound size reduction. Resiquimod application increased the area score of wounds starting on day 3 of treatment, and reached a maximum of 8.5 on day 7. The scores of the control wounds did not exceed the starting scores, and started to decline on day 4 (Figure 2B). The difference in the overall scores of resiquimod-induced and control wounds was highly significant, with *p* < 0.001.

Resiquimod application resulted in an increase in the wound size by up to 16.7% of the actual measured size (Figure 2C). The area scores of resiquimod-induced wounds increased from the starting score of 7 and remained on a high value until day 11, after which they declined until the end of the experiment (Figure 2B). In comparison, the wound area scores of the control wounds started to decline after day 3.

### 3.2. Resiquimod Treatment Initiated Inflammation and Necrosis

The mean inflammation score of control wounds reached its maximum of 0.5 on day 4, and went down to baseline levels on day 6. The mean inflammation score of the resiquimod-induced wounds increased significantly after day 2, reaching a maximum of 2.3 on day 8, but starting to decline to control wound levels on day 10 (Figure 3A). The difference in the scores of the control wounds compared to the resiquimod-induced wounds was highly significant, with *p* < 0.001 between days 2 and 11.

In line with elevated wound area score and inflammation score, resiquimod significantly increased the overall wound score, which describes the sum of the inflammation score and the wound area score. In control wounds, the overall wound score decreased between wound infliction and the study’s end (day 16). In comparison, the wound score of resiquimod-induced wounds was higher than day 1 until day 11, and then it declined until day 16 (Figure 3A). The mean overall wound score in control wounds reached its maximum at 7.4 on day 4, while in resiquimod-induced wounds, it reached a maximum of 10.7 on day 8 of the study (Figure 3A). The difference in the overall wound scores of control wounds and the resiquimod-induced wounds was highly significant, with *p* < 0.001 between day 2 and the study’s end.

### 3.3. Resiquimod-Induced Wounds Promoted Increased Expression of IL6, MMP1, CD68, IL8 and COX-2 Expression and Led to Immune Cell Infiltration

To study the effects of resiquimod application on wound healing and inflammation in more detail, mRNA expression of inflammation and regeneration markers was analysed. The mRNA expression of CD68, COX-2, IL10, IL8, IL6, MMP1, TGFb, and INFg was examined on day 1, day 7 and day 16 (Figure 4). CD68, IL8, IL10, IL6 and MMP1 mRNA abundance increased in both control and resiquimod-induced wounds on day 7 and day 16 compared to day 1. Expression of the cytokine IL6 was significantly increased on day 7 and day 16 in resiquimod-induced wounds compared to controls. The expression of the macrophage marker CD68, the cytokine IL8 and the prostaglandin synthase/cyclooxygenase-2 COX-2 showed trends towards increased expression on day 7 in resiquimod-induced wounds compared to control wounds, but reached expression levels of control wounds on day 16. TGFb and INFg did not show increased expression over time and resiquimod application had no effect on expression levels. Expression of the matrix-metalloprotease MMP1 was increased on day 16 in resiquimod-induced wounds compared to control wounds.

Histological assessment of haematoxylin- and eosin-stained wound sections revealed that most control wounds were fully healed and showed a restored and even extended epidermal layer. In a few control wounds, a residual scab was still visible. Overall, control wounds showed a physiological epidermis and dermis arrangement, with leukocytes still residing and an adjacent unchanged subcutaneous adipose tissue layer (Figure 5A,B). In contrast, resiquimod-induced wounds showed prominent necrotic scabs consisting of thick collagen fibres, leukocytes and cell debris. The epidermal layer was not distinguishable from the dermis, and, within the scabs and dermis, adipocyte clusters were observed. (Figure 5C,D). In line with increased expression of inflammatory markers, such as CD68 and IL6, histological analysis on day 16 showed a prominent immune cell infiltration in resiquimod-induced wounds, which was also reflected by immune cell quantification. Lymphocyte and neutrophil counts were significantly increased in resiquimod-induced wounds (Figure 6).

### 3.4. Open Flow Microperfusion (OFM) Showed That Short-Term Resiquimod Application Promoted Cytokine Release into the Dermis

Short-term resiquimod application over 48 h caused a significant increase in cytokine release into the dermal interstitial fluid (dISF). In line with mRNA expression results from long-term treatment, IL8 and IL6 release was significantly increased upon resiquimod application. Additionally, TNFalpha and IL1beta were highly abundant in the dISF of resiquimod-induced wounds compared to non-wounded control areas (Figure 7).

## 4. Discussion

Prolonged inflammation is a major cause of chronic wounds on a cellular level. In chronic wounds, the organism is not able to resolve the physiological inflammation as part of the wound healing process, and transition to the proliferation phase is blocked [3]. Underlying mechanisms of a prolonged inflammation phase and delayed or disturbed transition into the proliferation phase in chronic wounds are still under investigation. Many factors have been associated with prolonged inflammation resulting in delayed wound healing [6]. In this study, we analysed the specific impact of inflammation on wound healing progression. In order to induce inflammation in a standardised manner, topical resiquimod application was performed to generate wounds with prolonged inflammation. Inflammation was documented by visual wound scoring, measuring mRNA expression levels of inflammatory mediators and histological assessment.

In our experimental setup, resiquimod application resulted in wounds with a delayed healing time compared to control wounds. The difference in wound size reduction was one of the major observations in this study. Resiquimod-induced wounds showed an increase in wound size of up to 5 mm in length due to the treatment, resulting in prolonged inflammation; no wound contracture was observed. The resiquimod-mediated pro-inflammatory stimulus for 6 days after wounding resulted in prolonged wound healing in comparison to physiologically healing wounds. This observation underlines the importance of prolonged inflammation in the development of chronic wounds, and should always be considered in further chronic wound research or model development.

The histological results, as well as the mRNA expression analysis, showed an increase in inflammation on a cellular and molecular level in resiquimod-induced wounds. Studies have shown that the TLR pathway plays an important role in wound healing, as it modifies inflammation in the healing process [18,20,26]. Overexpression of a TLR can lead to prolonged inflammation and wound healing time [19]. When using a substance that activates these inflammatory pathways, i.e., resiquimod activating TLR7/8, no additional damage was done to tissue and cells compared to, for example, an irradiation model. Overexpression of TLR7 activated dendritic cells, suppressed anti-inflammatory T cells and further induced the expression of type I interferons, resulting in an enhanced inflammation [18,19].

Lipopolysaccharide (LPS) is commonly used in experimental models to induce inflammation [6,27]. While LPS can mimic wound infection with Gram-negative bacteria, such as *Pseudomonas aeruginosa* [27], infections with Gram-positive bacteria are not adequately reflected, although *Staphylococcus aureus* is commonly found in chronic wounds [28]. Moreover, LPS composition is dependent on the species and strains of the bacteria, from which it is isolated. The composition of LPS determines its binding affinities to TLR4 and/or TLR2, and therefore its inflammatory capacities [18]. Resiquimod, a chemically synthesised substance, specifically binds to TLR7/8, and has been successfully used in experimental models for inflammatory skin pathologies, e.g., psoriasis models [29]. By using resiquimod as the inflammatory agent, we expected more comparable results and fewer inter-individual variations. Further, in vitro data demonstrated that LPS or resiquimod induction of monocytes resulted in similar outcomes regarding immune cell priming and cytokine production [30].

Expressions of CD68, IL8 and COX-2 were increased in resiquimod-induced wounds only on day 7 of treatment, despite a strong immune cell infiltration seen on day 16, based on histological data. However, IL6 was significantly increased on both day 7 and day 16 in resiquimod treated wounds, indicating differences in the initiation and termination of specific inflammatory pathways. MMP1 expression, however, was first slightly decreased on day 7, but then significantly increased in resiquimod-induced wounds on day 16, when compared to untreated wounds. Comparable results have been obtained in a rat model of delayed wound healing, where MMP1 expression also increased in the delayed wound healing phase [31]. In patients, wound fluid of chronic wounds showed a marked increase in MMP1 protein abundance [32]. Furthermore, a previous study suggested that MMP1 and MMP2 could serve as good predictors of healing probability in chronic wounds [33]. They also found that COX-2 expression did not correlate with healing probability, which is also partly true for our study, where we only found minor changes in COX-2 expression in delayed wound healing [33].

In this study, we were able to show that resiquimod induced prolonged inflammation and significantly delayed wound healing. Since resiquimod application resembles wound phenotypes of prolonged inflammation that are comparable to patients with chronic wounds [32], controlled resiquimod application could be used to mimic chronic wounds in pigs and serve as a model for delayed wound healing. For example, high levels of CD68 expression have been observed in diabetic foot ulcers [34], and leucocyte infiltration has been reported in chronic wounds [35]. Although our findings indicate that the wound model with prolonged inflammation is comparable to human wounds with prolonged inflammation, more in-depth characterisation is still needed. For example, it must be clarified whether prolonged inflammation interferes with wound perfusion and angiogenesis, which are also features of chronic wound development [36,37]. Non-invasive in vivo imaging approaches, such as hyperspectral or thermal imaging, could be used to assess the impact of prolonged inflammation on wound perfusion [38,39]. Additionally, thorough immune cell phenotyping could unravel the involvement and activation of different types of immune cells, such as mast cells, in the course of impaired wound healing [40]. Furthermore, new factors involved in impaired wound healing need to be investigated, since a recent study also provides insights into the involvement of miRNAs [41]. miRNA-497, for example, exerts anti-inflammatory properties, which can counteract accelerated inflammation in chronic wounds [41]. This model should not serve as a substitute for other chronic wound models, but could act as basis to establish a multi-modal chronic wound model in pigs that more accurately reflects chronic wounds in humans.

## 5. Conclusions

By using resiquimod, we were able to study prolonged wound inflammation and delayed wound healing in a highly standardised way. Some limitations have to be mentioned. We used young immunocompetent pigs with no induced primary diagnoses, such as diabetes mellitus. Therefore, the delay in wound healing could be evaluated, but no direct comparison could be made relative to chronic wounds that develop after a long history of primary diseases. Further, we observed high inter- and intra-individual variation in expression patterns. Biopsies for mRNA expression analysis were taken on day 7 and day 16 from the outer parts of the wounds, the wound edges, whereas the histological samples were taken from the centres. We believe that the area where the biopsies were taken influenced the difference in cytokine expression and histology, since the centres of the wounds were most exposed to the resiquimod application and healed the slowest, and epithelialisation progresses from the wound edges to the wound centre [3]. Since differences in wound edge and wound base can be expected, serial biopsies from both areas need to be compared in future studies using this model of prolonged inflammation.

Since results from molecular and histological analysis were similar to human non-healing wounds with prolonged inflammation, we believe that this model builds a promising basis for further mechanistic studies of the effects of inflammation on chronic wound development.

## Figures and Tables

**Figure 1 biomedicines-10-00856-f001:**
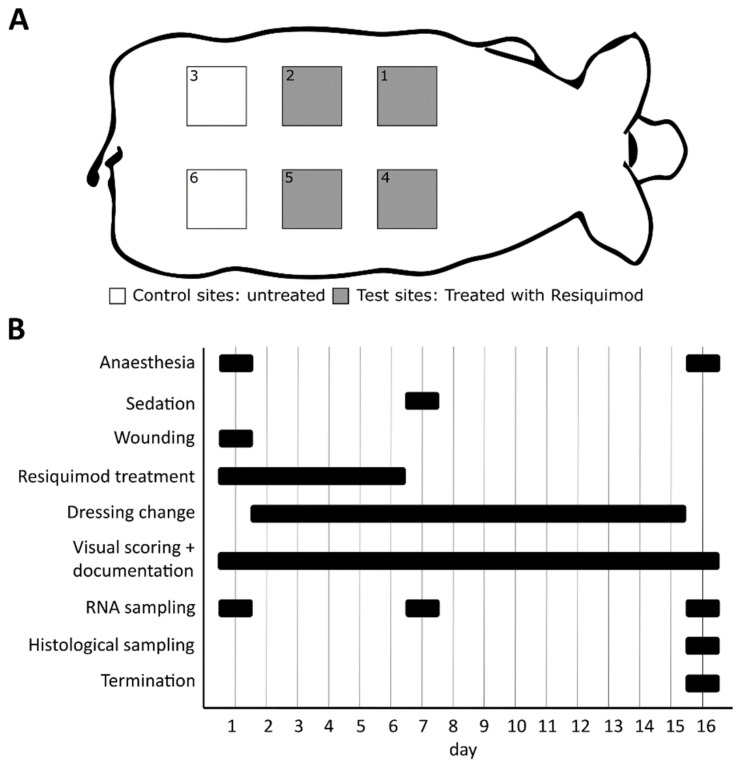
Schematic presentation of the experiment setup. (**A**) Each pig (n = 10) received 6 wounds on its back, 4 of which were treated with resiquimod, and 2 of which remained untreated as control wounds. Control and resiquimod-induced wounds were randomly arranged on the back of each pig to avoid a location bias. (**B**) The entire duration of the experiment was 16 days, during which the animals were anesthetised twice and received a sedation once. Resiquimod application lasted for 6 days. Dressing changes were performed on a daily basis, as were visual scoring and documentation. Biopsies for gene expression analysis were collected on days 1, 7 and 16, and biopsies for histology were collected only on day 16.

**Figure 2 biomedicines-10-00856-f002:**
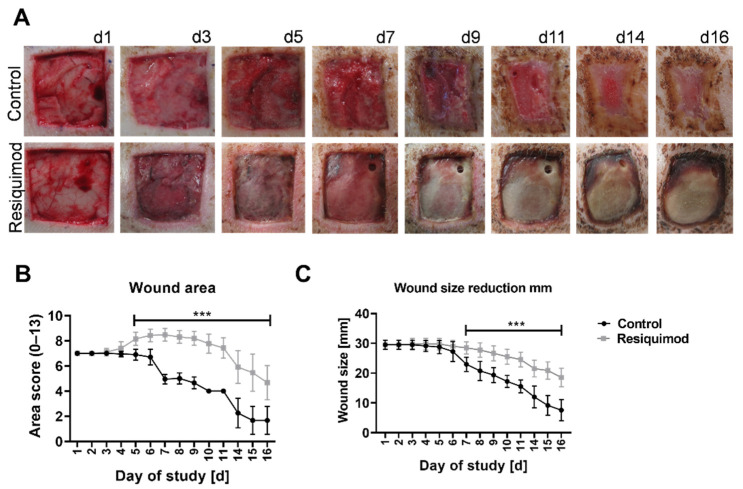
Wound morphology and wound area over the course of the experiment. Wounds of 3 × 3 cm were inflicted on the backs of pigs. Resiquimod was applied for 6 days, and the wounds were further examined over the course of 10 days. (**A**) Daily visual documentation and scoring was done. The control wounds ((**A**)–top row) showed physiological wound healing as well as wound contracture. The resiquimod-induced wounds ((**A**)–bottom row) showed a delay in wound healing and the formation of necrotic scabs. (**B**) The area scores showed higher values in resiquimod-induced wounds. (**C**) The wound size was determined by measuring the width of the wound, and less wound size reduction was shown in resiquimod-induced wounds. Data are presented as mean (dot) ± SD (whiskers) (n = 20 for control wounds; n = 40 for resiquimod-induced wounds). Data shown as mean ± SD. Statistical significance has been determined by 2-way ANOVA and corrected for multiple comparisons using the Sidak method. *** (*p* < 0.001).

**Figure 3 biomedicines-10-00856-f003:**
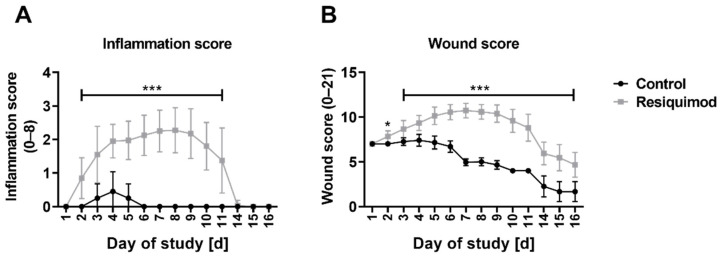
Inflammation score and wound score over the course of the experiment. (**A**) The inflammation score showed a highly significant difference between control and resiquimod-induced wounds. (**B**) The overall wound score includes the area score as well as the inflammation score. Data are presented as mean (dot) ± standard deviation (whiskers) (n = 20 for control wounds; n = 40 for resiquimod-induced wounds). Data shown as mean ± SD. Statistical significance has been determined by 2-way ANOVA and corrected for multiple comparisons using the Sidak method. *** (*p* < 0.001).

**Figure 4 biomedicines-10-00856-f004:**
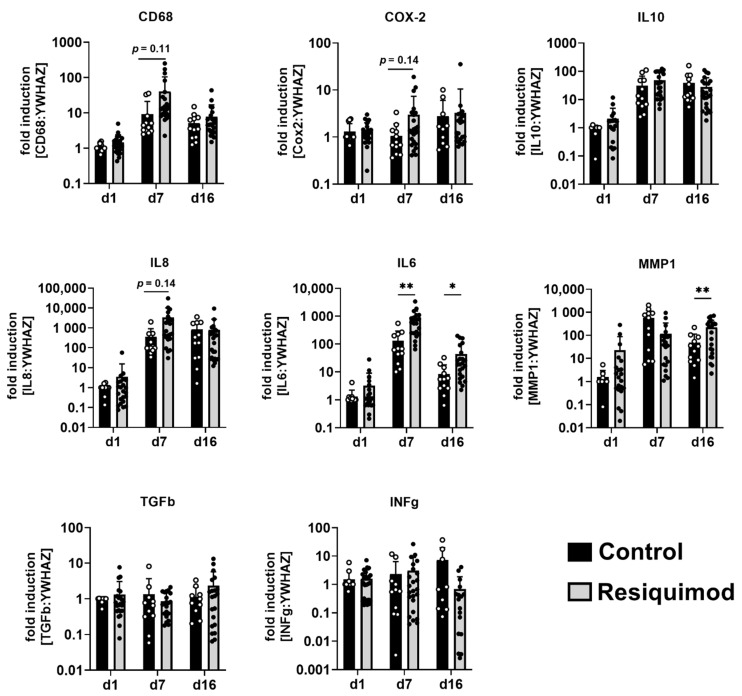
Resiquimod-induced inflammation increased the expression of IL6, MMP1, CD68, IL8 and COX-2. Relative mRNA expression levels of CD68, COX-2, IL10, IL8, IL6, MMP1, TGFb and INFg were determined by qPCR, normalising target gene expression to the averaged expression of YWHAZ. Resiquimod-induced wounds (grey) were compared to control wounds (black). Data are presented as mean (bar) + standard deviation (whiskers) (n = 9–12 for controls and 17–24 for resiquimod application). Data shown as mean + SD. Statistical significance has been determined by 2-way ANOVA and corrected for multiple comparisons using the Sidak method. * (*p* < 0.05), ** (*p* < 0.01).

**Figure 5 biomedicines-10-00856-f005:**
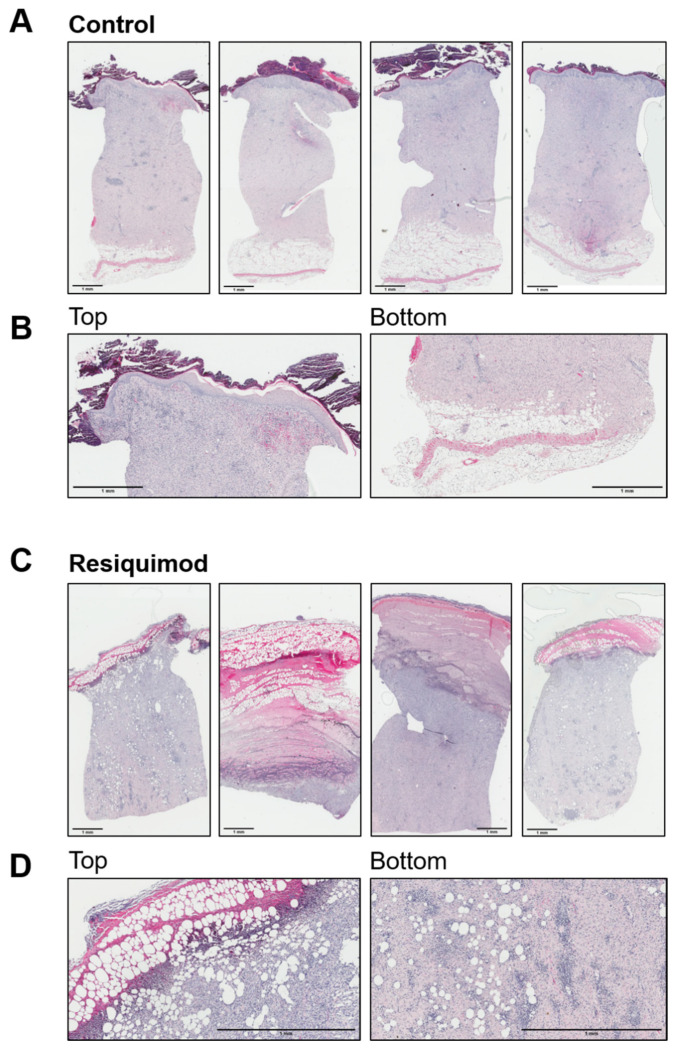
Resiquimod-induced inflammation leads to increased immune cell infiltration. (**A**) Four representative full overviews of haematoxylin/eosin staining of control wounds. (**B**) Detailed overview of one representative picture (left-top of the section); stratum corneum, epidermis and dermis of control wounds (right-bottom of the section) and dermis and subcutaneous adipose tissue of control wounds. Control wounds show physiological layers of skin, including a scab on top of the epidermis and a physiological layer of subcutaneous adipose tissue. (**C**) Four representative full overviews of haematoxylin/eosin staining of resiquimod-induced wounds. (**D**) Detailed overview of one representative picture (left-top of the section) of a barely existing stratum corneum, fully altered epidermis and dermis of resiquimod-induced wounds and (right-bottom of the section) a dermis with dispersed adipocytes in resiquimod-induced wounds. Resiquimod-induced wounds show high immune cell infiltration and no visible skin layers except the dermis. Scale bar = 1 mm.

**Figure 6 biomedicines-10-00856-f006:**
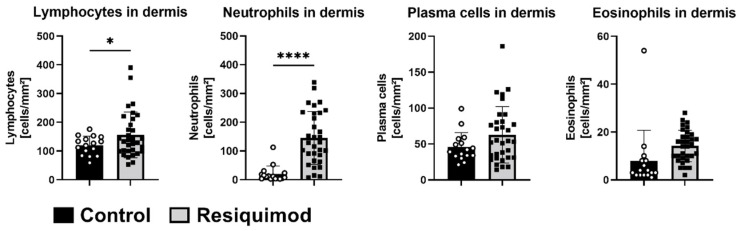
Resiquimod-induced inflammation increases lymphocyte and neutrophil infiltration. Immune cells were counted in haematoxylin- and eosin-stained sections from biopsies at the study’s end (day 16). Lymphocytes and neutrophils were significantly increased in the dermises of resiquimod-induced wounds, whereas no differences in plasma cell or eosinophil counts were detected (n = 16 for control wounds and 32 for resiquimod-induced wounds). Data shown as mean ± SD. Statistical significance has been determined by unpaired t-test with a Welch’s correction. * (*p* < 0.05), **** (*p* < 0.0001).

**Figure 7 biomedicines-10-00856-f007:**
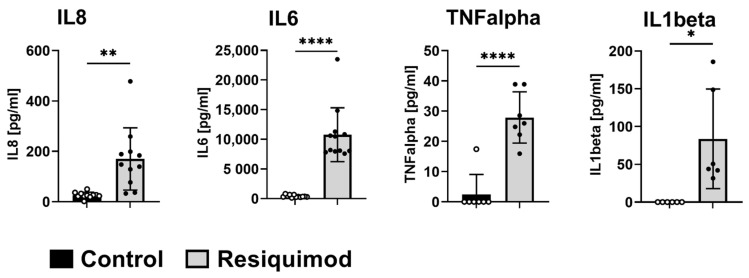
Short-term resiquimod application promotes the release of cytokines in the dermis. Release of cytokines into dISF was analysed by OFM. Split-thickness wounds were inflicted and treated with resiquimod and, 48 h later, OFM probes were implanted to sample dISF. Total IL8, IL6, TNFalpha and IL1b content was measured (n = 6–12 per treatment). Data are shown as mean ± SD. Statistical significance was determined by unpaired t-test with a Welch’s correction. * (*p* < 0.05), ** (*p* < 0.01), **** (*p* < 0.0001).

## Data Availability

The datasets generated and analysed during the current study are available from the corresponding author on reasonable request. All data generated and analysed during this study are included in this published article (and its Appendix A).

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
