# Peer review of "The Impact of Prolonged Inflammation on Wound Healing"

_biomedicines, 2022, doi:10.3390/biomedicines10040856_

Round 1

Reviewer 1 Report

In this article, Holzer-Geissler et al. described an elegant model on large animal (pigs) to mimic inflammatory skin disorders and chronic wound.

They apply a full thickness wounds on the back of pig and then treat with topical resiquimod application in order to provoke an inflammatory response. Authors clearly show that resiquimod induced prolonged inflammation and significantly delayed wound healing. The study is well presented and supported by consistent experiments that bearing the hypothesis.

Specific points:

  • Authors should discuss why they select only female pig for this study
  • Page 7 line 257, “11” instead of “14”?
  • Page 7 line 262, the wound score is not “increased” but rather “higher than day 0”
  • An important point will be a molecular study on TLR7 pathway (immunohistology or WB analysis for example). This should be added since authors highlighted the putative central role of this protein in their model.
  • Page 11 line 387, authors claim “MMP1 expression however, was first slightly decreased on day 7 but then significantly increased in resiquimod-induced wounds at day 16”. Authors should add “comparing to untreated wound”.
  • The major point to be addressed is that authors should present an immunostaining on histology at day 16 on MMP1 for example and should compare with a human wound histology with the same marker(s) to support the animal model and the use for human preclinical applications.

Reviewer 2 Report

The manuscript presented by Judith CJ Holzer-Geissler et al. titled: “The impact of prolonged inflammation on wound healing” is well written, clear, and easy to read. The topic is very interesting and therefore, it adds clustered information to the subject area of wound healing mediated by a chronic condition.

In particular, using resiquimod the scientists thought to a very nice model to study this skin chronic condition with peculiar pathophysiological aspect, also in pigs the animal model with the immune system more similar to human.  This highlights how it is important to consider molecule/drug which besides antibacterial activity should have also anti-inflammatory activity.  

In the discussion section please add this two reference:

  • Biomed Pharmacother. 2020;121:109613. doi:10.1016/j.biopha.2019.109613
  • Int Wound J. 2020;17(2):485-490. doi:10.1111/iwj.13299.

For me, it can be accepted after this minor revision of the present form.
